# Comparison of Recruitment Patterns of Sessile Marine Invertebrates According to Substrate Characteristics

**DOI:** 10.3390/ijerph19031083

**Published:** 2022-01-19

**Authors:** Seongjun Bae, Michael Dadole Ubagan, Sook Shin, Dong Gun Kim

**Affiliations:** 1Department of Ecology and Conservation, Marine Biodiversity Institute of Korea, Seocheon 33662, Korea; silverto@mabik.re.kr; 2Department of Ocean Environmental Sciences, College of Natural Science, Chungnam National University, Daejeon 34134, Korea; 3Department of Animal Biotechnology and Resource, College of Science and Technology, Sahmyook University, Seoul 01795, Korea; summer_09.breeze@yahoo.com; 4Smith Liberal Arts College, Sahmyook University, Seoul 01795, Korea

**Keywords:** artificial substrate, sessile invertebrates, marine biofouling, colonial ascidian, bivalve, Cirripedia

## Abstract

A community of benthic invertebrates, including sessile adult-stage invertebrates, can negatively effect corrosion, deformation, and increased fuel consumption by attaching to artificial structures, a phenomenon known as marine biofouling. Investigating the relationship between benthic communities and artificial structures or substrates (to which the organisms attach) can help clarify the factors influencing marine biofouling. Therefore, in our study, natural (stone) and artificial (rubber, tarpaulin, and iron) substrates were installed in three harbors (Mokpo, Tongyeong, and Busan), and the structures of the communities attached to each substrate were compared. The total study period was 15 months (September 2016 to December 2017), and field surveys were performed at 3-month intervals. The three survey sites had significant differences in the structure of the sessile community present. In particular, Tongyeong was significantly different from Mokpo and Busan due to the continuous dominance of Cirripedia. When comparing natural and artificial substrate by sites, significant differences were observed in the community structure in all three surveyed sites. In Mokpo and Busan, colonial ascidians were dominant on natural substrate rather than artificial substrates; post-summer, Cirripedia coverage was higher on artificial substrates than natural substrate due to corrosion. Tongyeong showed a different pattern from that of Mokpo and Busan. After the summer, Bivalvia dominated on natural substrate over artificial substrates, affecting the differences between natural and artificial substrates. Our results demonstrate the recruitment patterns of sessile marine invertebrates according to substrate characteristics and can be used as basic information for biofouling management in marine environment.

## 1. Introduction

Many benthic taxa attach to and inhabit a variety of natural substrates, such as mollusk shells, algae, and rocks. The life cycle of sessile benthic invertebrates is generally biphasic and divided into planktonic and sessile adult stages [1,2]. Floating larvae live for a few minutes to several months before attaching to a substrate surface. On encountering the appropriate substrate, they settle and develop into juveniles [2]. Larval attachment is affected by several physical and chemical environmental variables, such as water temperature, depth, food access, substrate surface, and microtopography [3,4,5,6,7,8,9]. The abundance of sessile benthic invertebrates can be greatly affected by natural and artificial structures (e.g., the bottom of a ship, aquaculture facilities, and power plants) [10,11], and these surface differences lead to changes in the community composition [2,12,13]. Furthermore, artificial structures can be negatively impacted as a result of corrosion, deformation, and increased fuel consumption by sessile benthic invertebrate communities [14,15], through the phenomenon known as marine biofouling.

Biofouling is undermining the primary purpose of marine structures globally. Biofouling of ship damages the structure of the hull through corrosion and deformation. In addition, ship consume additional fuel due to the increased weight and the greater drag effect [16,17]. Increasing fuel consumption exacerbates CO_2_ emissions and incurs higher costs [18]. Therefore, to offset the economic threat, biofouling management and control are critical. Such management, implemented regularly, can also help stabilize the health of marine ecosystems. Thus, understanding which artificial structures or substrates organisms preferentially attach to is a critical factor in understanding marine biofouling [19,20] because the structure of benthic invertebrate communities change according to the type of artificial structure [8,21,22,23].

The impact of marine biofouling on artificial structures was not considered significant until the 1980s [24]. However, as the development of ports has accelerated with the growth of cities worldwide [25,26], there physical, chemical, and biological environments have likewise changed with the increased use of artificial structures; biofouling of such structures has caused functional and financial problems [27,28]. As a result, various benthic invertebrate studies using artificial structures have been conducted [29,30,31]. These include investigation of the attachment and development of benthic invertebrates in function of the location and type of marine installations [32] and the analysis of changes in communities living on artificial structures [33]. Recently, various studies have been conducted using artificial substrates, such as differences in substrates, changes in group structure of benthic invertebrate communities, and differences between vertical and horizontal substrates [34,35,36]. Artificial substrates have the advantage of being easy to design and replicate and are relatively easy to handle in the field [34,37,38,39]. Therefore, various studies on marine biofouling have been conducted on artificial substrates. These include substrate types (polyvinyl chloride [PVC], ceramic tile, and stone), directions, and the installation environment [34,35,36,40,41,42].

Although some researchers have compared different types of artificial substrates [43,44,45,46], few have compared a variety of artificial substrates with natural substrates. Further research is needed, and we are interested in elucidating the differences between natural and artificial substrates. Therefore, the purpose of this study was to determine how artificial substrates affect the recruitment, abundance, and diversity of marine benthic invertebrates compared to natural substrates in several harbor habitats. Specifically, we tested the following: (1) differences in the community structure of natural substrates due to port location; (2) differences between natural and artificial substrates due to the community structure inhabiting the substrate; and (3) the effects of differences among artificial substrates on species composition and abundance.

## 2. Materials and Methods

### 2.1. Experimental Design

This study was conducted in three regions across two provinces (Jeollanam-do and Gyeongsangnam-do) in the southern part of the Korean Peninsula, and three research sites with different sea area characteristics. Specifically, the sites were the Mokpo Yacht Marina (34°47′2.70″ N, 126°23′21.05″ E) in Mokpo City located in Jeollanam-do (hereafter MP), the Marine Sports Center (34°49′38.80″ N, 128°26′4.22″ E) in Tongyeong City located in Gyeongsangnam-do (hereafter TY), and Busan Port Passenger Terminal (35°5′56.41″ N, 129°2′19.40″ E) in Busan City located in Gyeongsangnam-do (hereafter BS). MP is in the Yellow Sea area affected by the Korean coastal current; TY and BS are in the Korean Strait affected by the Tsushima warm current. MP is affected by the Korean coastal current flowing from the north during winter, and the coastal currents are weakened by the influence of monsoon winds in summer [47,48]. The Kurushio current, which affects TY, provides warm water in summer and is highly salinity. [49,50,51,52]. BS is affected by the Tsushima warm current and the freshwater discharge of the Nakdong River [52]. The selected study sites are places where yachts and ships are loaded in the harbor, and various artificial structures, such as port walls and floating docks, exist. Three points were selected (for replicates) on each study site, and artificial substrates were installed at a depth of 3–4 m (Figure 1).

One natural substrate (stone) and three artificial substrates (rubber, tarpaulin, and iron) were used in the experiment. Stone is a typical substrate in rocky shore ecosystems and a major habitat and recruitment substrate for benthic organisms. It occurs extensively along the coastline and consists of hard materials, such as granite, chalk, basalt, and limestone [53]. In this study, basalt was selected as the natural substrate. Rubber is the primary material used for the tires installed on ships or harbor walls to prevent impact, and tarpaulin is a fabric commonly used for buoys and fishing. Iron is the most representative substrate on boats and is also used in other artificial structures, yet is rarely used because it is coated with paints and other materials. All substrates were prepared in the same form, as 300 × 300 mm squares. The stone was prepared by cutting a basalt plate (300 × 300 × 12 mm). For the rubber, a sheet for engraving was cut (300 × 300 × 4 mm), and the tarpaulin was covered with an acrylic plate (300 × 300 × 3 mm). The iron was made using alloy steel (300 × 300 × 3 mm), used for making ships and various artificial structures.

The study duration was around 15 months, from September 2016 to December 2017, and five field surveys were performed at 3-month intervals. To reduce disturbance to the organisms attached to each substrate during the field survey, the substrate plate was quickly lifted and placed in a plastic box (665 × 330 × 195 mm) filled with local seawater. After recording various factors of substrate in the box, it was restored to the original position. Biological factors (species richness and coverage) were recorded in the survey log and photographed using a camera (Tough TG-5, Olympus, Japan) to estimate species recruitment and species composition changes over time. In addition, the environmental factors (air temperature, water temperature, salinity, electrical conductivity, dissolved oxygen, and hydrogen ion index) were recorded using a multi-sensor meter (Pro Plus, YSI, USA) to identify changes in physical environmental factors. For continuous water temperature and salinity information, data provided by the Korea Hydrographic and Oceanographic Agency of Ocean Data in Grid Service (https://www.khoa.go.kr; accessed on 5 August 2021) were used. In addition, for precipitation data, observation results provided by the Korea Meteorological Administration Open MET Data Portal (https://data.kma.go.kr/; accessed on 15 October 2021) were used.

### 2.2. Data Analyses

Species identification was performed in the field. Regarding species with small individuals or taxa that were difficult to identify, the encyclopedia and various literature were referenced [53,54,55,56,57,58,59] based on photographic data. Each taxon was analyzed by grouping it at the phylum or class level for statistical analysis. Bryozoans and ascidians were divided into two groups according to the morphotype attached to the substrate (Appendix A). Bryozoans were categorized as either branching or encrusting, according to the architecture and style of disarticulation or fragmentation. Branching organisms grow vertically upright and the entire colony is flexible; in contrast, encrusting organisms form single-or multi-layered hard calcareous structures [60,61,62]. Ascidians were also morphologically and ecologically categorized as either solitary or colonial. Solitary ascidians are independent individuals and reproduce exclusively by sexual reproduction, whereas colonial ascidians form colonies and reproduce sexually and asexually by budding and strobilation (Appendix A) [63,64].

The averages over the three replicates per site of species richness and coverage (percent cover) were used for all data analyses. Coverage was calculated using the grid cell counting method in photoQuad, version 1.4 (V. Trygonis & M. Sini., Univ. of the Aegean, Mytilene, Lesvos island, Greece) [65], based on photographic data in the laboratory. Permutational multivariate analysis of variance (PERMANOVA) was run on a triangular similarity matrix derived from the Bray–Curtis dissimilarity index of square-root transformed data. PERMANOVA was used to verify the significance of three sites and four substrates. The permutational analysis of multivariate dispersions (PERMDISP) was used for testing the homogeneity of multivariate dispersions from group centroids on the basis of the resemblance measure. Non-metric multidimensional scaling (nMDS) was used to analyze statistical differences at each study site and in the community of natural substrates. Rare species (i.e., those with a frequency of <5%) were excluded from the analysis, and the data were relativized for coverage of species to the same portion [66]. We used multi-response permutation procedures (MRPP), a nonparametric protocol, to test the null hypothesis of no differences among groups in the data matrix and determine the statistical significance of each study site. Analysis of similarities (ANOSIM) was performed to measure the dissimilarity of the substrate type. This was based on the Bray–Curtis similarity index with 9999 permutations [66]. In the ANOSIM result, it was considered statistically significant when the *p* value was less than 0.05 and the R value was greater than 0.5. The similarity percentage analysis (SIMPER) was performed to compare the coverage of taxa between the communities and identified which taxa characterized the different substrate. One-way analyses of variance (ANOVA) were conducted to compare the mean number of species and coverage of benthic invertebrates using the Tukey–Kramer method. Statistical significance was set at *p* < 0.05. A *p*-value from two-way ANOVA was adjusted by the Benjamini–Hochberg procedure to control for false-positive discoveries in the positive dependent assumption [67], and an adjusted *p*-value of less than 0.05 was considered statistically significant (Appendix A). PERMANOVA, PERMDISP, and ANOSIM were conducted using PRIMER-e, version 6 (Quest Research Ltd., Auckland, New Zealand) with the PERMANOVA+ add-on package [68,69]. In addition, nMDS, SIMPER, and MRPP analyses were conducted using the PC ORD software, version 6 [70]. ANOVA was performed using SPSS Statistics for Windows, version 22.0 (IBM Corp., Armonk, NY, USA).

## 3. Results

### 3.1. Environment and Fauna

The study period lasted about 15 months, from September 2016 to December 2017, and a field survey was conducted every 3 months for about 470 Julian days. During this period, various environmental factors were observed. The water temperature of MP observed at the site during the survey period ranged between 4.1 and 27.6 °C, and the average water temperature was 14.9 ± 7.1 °C (mean ± SD). The water temperature in TY ranged between 5.3 and 30.4 °C, and the average water temperature was 15.9 ± 6.9 °C. In the case of BS, the water temperature ranged between 9.5 and 28.4 °C, and the average water temperature was 16.5 ± 4.7 °C. The observed salinity in MP ranged between 21.8 and 33.3 PSU, and the average salinity was 30.4 ± 3.2 PSU. The salinity in TY ranged between 29.6 and 34.8 PSU, and the average salinity was 32.8 ± 1.3 PSU. In BS, salinity was between 27.7 and 34.3 PSU, and the average salinity was 33.2 ± 1.1 PSU (Figure 2). There were statistically significant differences in water temperature (F_2,1317_ = 7.74, *p* < 0.05) and salinity (F_2,1314_ = 301.43, *p* < 0.05) at each site. The total precipitation observed during the study period was the highest in TY at 1142.2 mm, and was 942.8 and 455.4 mm in BS and MP, respectively. In all surveyed sites, the precipitation of 2017 was high in summer (June to September; MP = 30.2%, TY = 37.7%, and BS = 40.8%) compared to other seasons (Figure 2, Appendix A). The tide ranges differed among the three sites. The daily tide in MP ranged from 189.9 to 459.5 cm, with an average of 254.9 ± 20.1 cm. In TY, it ranged from 118.8 to 184.2 mm, with an average of 152.4 ± 12.7 mm. In the case of BS, the tide range was between 47.5 and 105.0 mm, with an average of 74.17 ± 10.5 mm. There was a significant difference in the tidal level of each region (F_3,56_ = 641.18, *p* < 0.05; Appendix A).

Field investigations during the study period revealed the occurrence of 32 species of marine benthic invertebrates belonging to 19 families, 12 orders, 7 classes, and 6 phyla. Rare species (i.e., those occurring with a frequency of <5%; six species) were excluded from the analyses (Appendix A). BS had the highest species richness of the three surveyed areas, with 23 species observed during the entire study period, and Cirripedia, including *Amphibalanus trigonus*, and Bivalvia, including *Mytilus galloprovincialis*, were the main taxa. In particular, Cirripedia was observed during all survey periods, and coverage was over 15%. Twenty-one species were observed in MP, and the dominant taxa were Bivalvia, including *M. galloprovincialis*, and colonial ascidians, including *Didemnum vexillum*. In the case of Bivalvia, coverage of about 60% was observed after 420 Julian Days. Twenty species were observed in TY, and the dominant taxa were Bivalvia, including *Magallana gigas*, and Cirripedia, including *A. trigonus* and *A. improvisus* (Figure 3). Up until Julian day 463, BS had significantly higher species richness than the other two sites (*p* < 0.05; Appendix A).

### 3.2. Spatial and Temporal Analyses

The PERMANOVA results indicated a significant difference between the two factors (three sites and four substrates) over the entire study period. Comparison of differences between sites was significant (*p* = 0.0001), and differences between substrates were not (*p* = 0.0615). There was no significant interaction between the two factors (site × substrate; *p* = 0.2253). In overall individual pairwise tests, only MP vs. TY (*p* = 0.000) and TY vs. BS (*p* = 0.004) had significant differences, and other factors did not show significant differences (*p* > 0.05). The overall PERMDISP for testing the homogeneity of dispersions revealed significant differences only MP vs. BS (*p* = 0.020), and stone vs. iron (*p* = 0.046). For all other factors, PERMDISP generated *p* values > 0.05. Therefore, there was a significant difference in the benthic communities at each site (except MP vs. BS), but no difference between each substrate (except stone vs. iron; Table 1 and Appendix A). The nMDS analysis was conducted with nine quantitative variables (H’: Shannon diversity index, D: Simpson dominance index, J’: Pielou evenness index, d: Margalef richness index, R: species richness, TC: total coverage, WT: water temperature, Sal: salinity, and pH) and two categorical variables (sites and times) based on the sessile benthic invertebrate communities. The variances of axes 1 and 2 were 41.8% and 38.1%, respectively, and axis 1 appeared to be divided by time, and was inferred to be affected by TC, pH, R, and WT (Figure 4). The MRPP analysis comparing the surveyed sites showed a significant difference between MP and TY (*p* = 0.00369) and TY and BS (*p* = 0.00979), but not between MP and BS (*p* = 0.36589; Appendix A).

A two-way ANOVA analysis was performed with time and substrates for each taxon to determine whether there was a difference in substrates over time for each survey site. The analyses showed that in MP, coverage of Cirripedia and colonial ascidians exhibited a significant difference according to substrates and time (*p* < 0.05). In TY, only Bivalvia was significantly different (*p* < 0.05). In BS, Cirripedia and Bivalvia were significantly different (*p* < 0.05). Except for Cirripedia, Bivalvia, and colonial ascidians, there was no difference in any other taxa in the study sites (Table 2).

### 3.3. Natural and Artificial Substrate Analysis

When the coverage by major taxa was compared among substrates at each site, there was no significant difference in the community structure between natural and artificial substrates until about 200 Julian days. There was a difference in the rubber substrate of MP because Cirripedia had a higher abundance than other substrates, but there was no similar trend in other sites (Table 2; Appendix A). The coverage of Cirripedia, Bivalvia, and colonial ascidians on artificial substrates was different from that on the natural substrates. Cirripedia coverage on artificial substrates showed a difference from that on the natural substrate after approximately 270 Julian days in MP and BS. The mean coverage of iron was higher than that of natural substrates, and the difference was statistically significant (F_3,5–6_ = 7.60–15.25, *p* < 0.05; F_3,6–8_ = 5.49–8.90, *p* < 0.05). Colonial ascidians also favored different substrates in MP and BS. In MP on 270–358 Julian days and in BS on 185 Julian days, the mean coverage of natural substrates was higher than the mean coverage of artificial substrates (F_3,7_ = 17.50–50.25, *p* < 0.05; F_3,7_ = 4.66, *p* < 0.05). The Bivalvia showed differences in substrate preference in TY. On 269–357 Julian days, the mean coverage of natural substrates was higher than that of artificial substrates (F_3,5–7_ = 7.50–20.34, *p* < 0.05). According to the ANOSIM results, there were statistically significant differences in stone vs. tarpaulin (R = 0.692, *p* = 0.008) and stone vs. iron (R = 0.521, *p* = 0.011) due to colonial ascidians in MP. There were significant differences in TY in stone vs. tarpaulin (R = 0.728, *p* = 0.007) and stone vs. iron (R = 0.869, *p* = 0.002) due to Bivalvia. In BS, there was a significant difference between stone and rubber (R = 0.517, *p* = 0.001) and stone vs. tarpaulin (R = 0.619, *p* = 0.005) due to Bivalvia, and there was a difference in stone vs. iron (R = 0.745, *p* = 0.011) due to Cirripedia (Table 3). SIMPER results also showed the same trend as ANOSIM results. In Mokpo, there was a difference between natural and artificial (rubber, tarpaulin, and iron) substrate due to the high contribution of colonial ascidians (>25%), and in Tongyeong, there was a difference between stone vs. tarpaulin and stone vs. iron due to the contribution of Bivalvia (>18%). In BS, there was a significant difference between stone vs. rubber and stone vs. tarpaulin due to Bivalvia (>19%), and there was a difference in stone vs. iron due to Cirripedia (>22%; Appendix A).

## 4. Discussion

### 4.1. Environment and Spatial Fauna Analysis

The spatial difference of the survey area affected by different currents also affects environmental variables, such as temperature and salinity. The mean water temperature and mean salinity were the lowest in MP and the highest in BS, but the mean tide level was the highest in MP and lowest in BS (Figure 2; Appendix A). There was a significant difference in the environmental factors (surface water temperature, salinity, and tide level) at each study site (*p* < 0.05), and this was considered to influence the differences in the community structures. The results of PERMANOVA and PERMDISP analysis also supported the differences in each site, but the differences in substrates were not significant (Table 1 and Appendix A). Therefore, the analysis of the substrates difference and the community structure was performed independently for each survey site.

Cirripedia, a dominant taxon in all survey sites is one of the most common macrofouling organisms in the marine invertebrate community [71,72]. TY had the highest abundance of Cirripedia (44.2%), and the most dominant genera were *Amphibalanus* and *Balanus*. Among them, *B*. *trigonus* is a common species in the Korea Strait [73], and in the cyprid stage, before becoming an adult, it tolerates a wide range of temperatures (20–30 °C) and salinity (10–45‰) [74,75]. *Balanus improvisus* is distributed in the Yellow Sea and Korea Strait, and *Amphibalanus eburneus* is distributed in the Korea Strait; these species tolerate low salinity [76,77,78]. These temperature and salinity ranges were consistent with those of TY during the summer. In addition, Cirripedia was observed throughout the survey period in Tongyeong (Figure 3), and the fact that the number of Cirripedia species is the most distributed in the Korea Strait [73] supports the result that TY had the highest number of Cirripedia species.

Colonial ascidians, one of the main dominant taxa, are globally distributed, and the genera *Didemnum, Botrylloides*, and *Botryllus* found in this study are known to inhabit a wide range of environmental conditions, including temperature and salinity [79,80,81]. Brunetti et al. [82] conducted an experiment on optimal growth conditions for *B. schlosseri* of the genus *Botryllus* at wide temperature (13–25 °C) and salinity (25–40‰) ranges. They found that the optimum temperature and salinity were 20 °C and 33‰, respectively, and the relatively high optimum salinity was consistent with the average salinity of BS. Therefore, it is considered that the community structure became similar in MP and BS, owing to the high dominance of colonial ascidians, including *D. vexillum* and *B. schlosseri*. The analysis results of this study (nMDS and MRPP) based on benthic organisms of natural substrates also support this reasoning (Figure 4; Appendix A).

### 4.2. Natural vs. Artificial Substrate

There have been several previous studies on sessile marine invertebrates using artificial substrates. In particular, a study comparing natural and artificial substrates was conducted, and Cacabelos et al. [43] used two types of artificial substrates (basalt and concrete) with rough surfaces. Sedano et al. [59] directly compared natural and artificial substrates, but only one type of artificial substrate was used (rip-raps). The present study was conducted to test whether there was a significant difference between various types of artificial and natural substrates. In our study, the Cirripedia dominated all the surveyed sites. Cirripedia abundance on artificial structures (usually fairly smooth) is known to be low [43] because the roughness of the substrate surface is one of the major factors that encourages recruitment [83,84]. However, our study results showed that the abundance of Cirripedia was higher on iron in MP and BS than on natural substrates. Significant differences began to appear in MP from September and in BS from June. Iron is an artificial substrate used in ships and offshore structures; various studies have been conducted on the effect of fouling organisms on iron. For example, adherents were found to induce local corrosion of steel carbon [85]. In particular, the bottom surfaces of dead Cirripedia induce iron corrosion [86]. More recently, it has been found that the bases of live Cirripedia can also lead to corrosion [87]. Because of these Cirripedia characteristics, corrosion occurs in the iron substrate about 250–300 Julian days after erosion, and the electrochemical activity of the steel surface due to corrosion directly inhibits the settlement of fouling organisms; settlement probability may be reduced through physical surface instability [88,89,90]. There are studies which suggest that the population of Cirripedia increases due to iron corrosion [91], but organisms excluding Cirripedia are considered to be difficult to attach in the post-summer season. Consequently, when antifouling management is carried out in areas where Cirripedia dominates over other taxa, specific physical and chemical marine practices must be considered rather than general antifouling methods (Appendix A).

There was a difference in the abundance of natural and artificial substrates for colonial ascidian after June in MP and after March in BS. Although few studies have investigated the mechanism of ascidian settlement [92,93], it appears that *Ciona intestinalis* and *Botrylloides violaceus* adhere to the substrate through protein-based mechanisms [2]. In addition, ascidian larvae tend to prefer rough rather than smooth surfaces for their settlement substrates. This is potentially advantageous, as it reduces the risk of the organism becoming detached from the substrate [2]. However, because the adhesion strength of ascidians is relatively weak compared to that of other sessile organisms [94], the properties of the substrate affecting the adhesion strength are important. Therefore, it appears that the peculiar properties of colonial ascidians contributed to their higher abundance in the natural substrate. The surface complexity of the natural substrate is not the only explanation for the ascidian abundance. Depending on the substrate type, other factors, such as space competition with other organisms [95] and mineralogy, including the presence of silica, have been found to influence substrate adhesion [96]; therefore, these factors should be considered in further studies.

TY showed a different pattern from that of the MP and BS. After June, the abundance of Bivalvia was higher on natural substrates than on artificial substrates, and there was no difference on the substrates of Cirripedia and colonial ascidians (Table 2). In general, the Korea Strait is affected by the Tsushima warm current and differs from other sea areas. According to Park et al. [51], the highest water temperature of TY in August 2015 was similar to that of MP, but the temperature of BS was lower (MP = 24.9 °C; TY = 25.3 °C; BS = 19.7 °C). However, in August 2017, during the present study, the highest water temperature of TY was observed to be about 2 °C higher than that of MP and BS (MP = 27.6 °C; TY = 30.4 °C; BS = 28.4 °C). It is thought that such high temperatures in summer are problematic for the genera *Botryllus*, *Botrylloides*, and *Didemnum*, which rarely survive above 25 °C [80,97,98]. However, the dominant taxon in TY, the bivalve (*Mytilus edulis*), is a species that can survive at temperatures as high as 30 °C [98]. Therefore, the high-temperature environment of TY would favor *M. edulis* over other taxa. In addition, mussels and oysters form dense sets of various topographies on soft and hard marine benthic substrates [99,100]. Mussels prefer a hard substrate when attaching [101,102], and oysters, in particular, have shown more recruitment on natural rocky shores than on artificial structures in previous studies [103]. These bivalves are major space occupants [104,105], forming dense layers that contribute significantly to ecosystem structure and stability [98,106,107]. Furthermore, these two- or three-dimensional layers are unaffected by the type of substrate and form shelters and habitats for numerous organisms [107,108]. Thus, the higher summer water temperature in TY provided favorable environmental conditions for bivalves, and the dense layers they formed probably reduced differences between substrate types by providing new habitats for various other organisms.

## 5. Conclusions

In conclusion, our study analyzed the differences in marine invertebrate communities inhabiting natural and artificial substrates at three sites. The differences in the biological, geographical, and physicochemical environments of the three study sites influenced the sessile marine invertebrate communities. However, due to the influence of the community structure of MP and BS showing similar trends, there were significant differences exhibited in TY. The results of this study demonstrate that the effects of natural and artificial substrates on colonization were different for each study site. In MP and BS, the significant difference was due to colonial ascidians; in TY, it was due to bivalves. The differences in MP and BS were the result of the recruitment characteristics of colonial ascidians. Conversely, differences at TY resulted from conditions favoring bivalves, as high summer temperatures enabled the formation of dense layers of bivalves. Of the artificial substrates, the abundance of benthic invertebrate in iron at MP and BS differed from that in other substrates owing to the effects of Cirripedia. It is thought that Cirripedia increases iron corrosion, forming an environment that is difficult for other taxa to colonize. Understanding the effects of substrate materials on the settlement of marine fouling organisms is important for marine ecology. Because the substrates of the structures were submerged in water longer than the duration of the study period, differences between the study sites suggest that a site-based approach may be best suited to optimize antifouling strategies. In addition, regarding future management and antifouling, if any sites reflect the settlement patterns observed in this study, they will potentially promote biofouling depending on the substrate.

## Figures and Tables

**Figure 1 ijerph-19-01083-f001:**
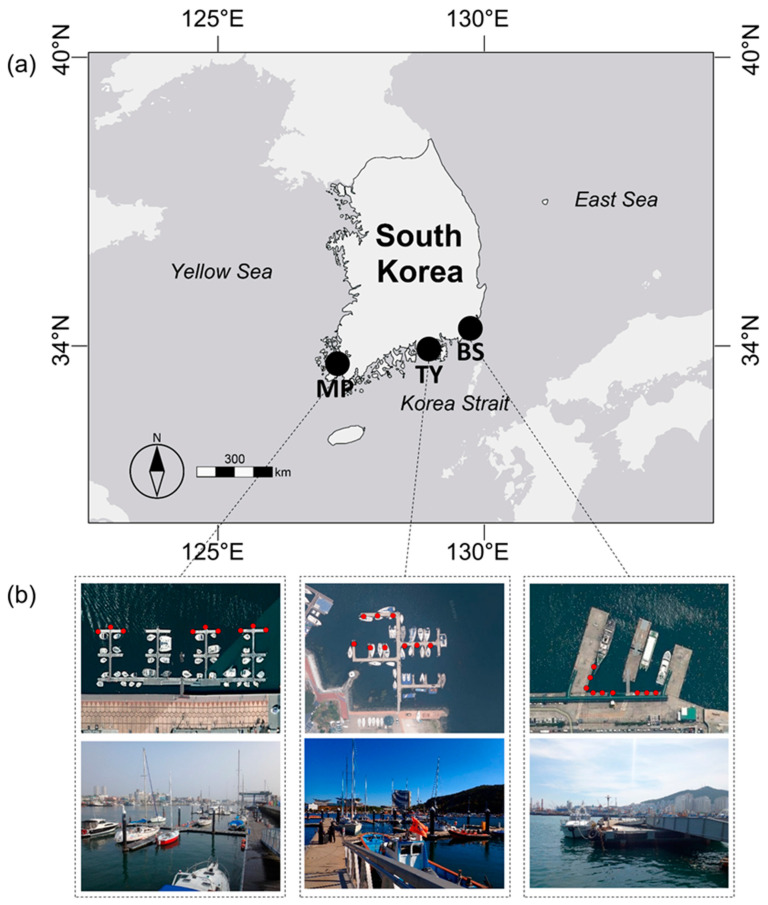
Map showing the location of the three survey sites in the southern part of the Korean Peninsula (**a**), and satellite image including the installation position (red dot) of the substrates and foreground photo in each survey site (**b**). MP, Mokpo; TY, Tongyeong; BS, Busan.

**Figure 2 ijerph-19-01083-f002:**
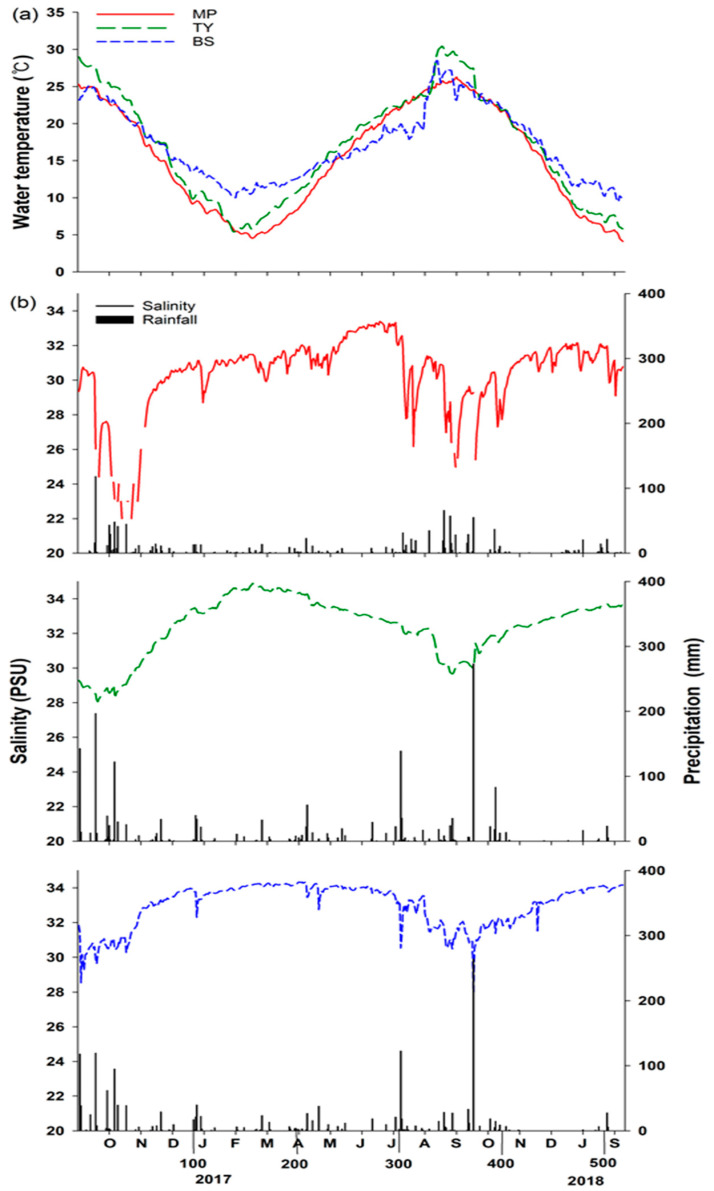
Daily average water temperature results (**a**) and daily average salinity (red line: MP, green line: TY, blue line: BS) and precipitation (vertical bar) results (**b**) for each site during the study period.

**Figure 3 ijerph-19-01083-f003:**
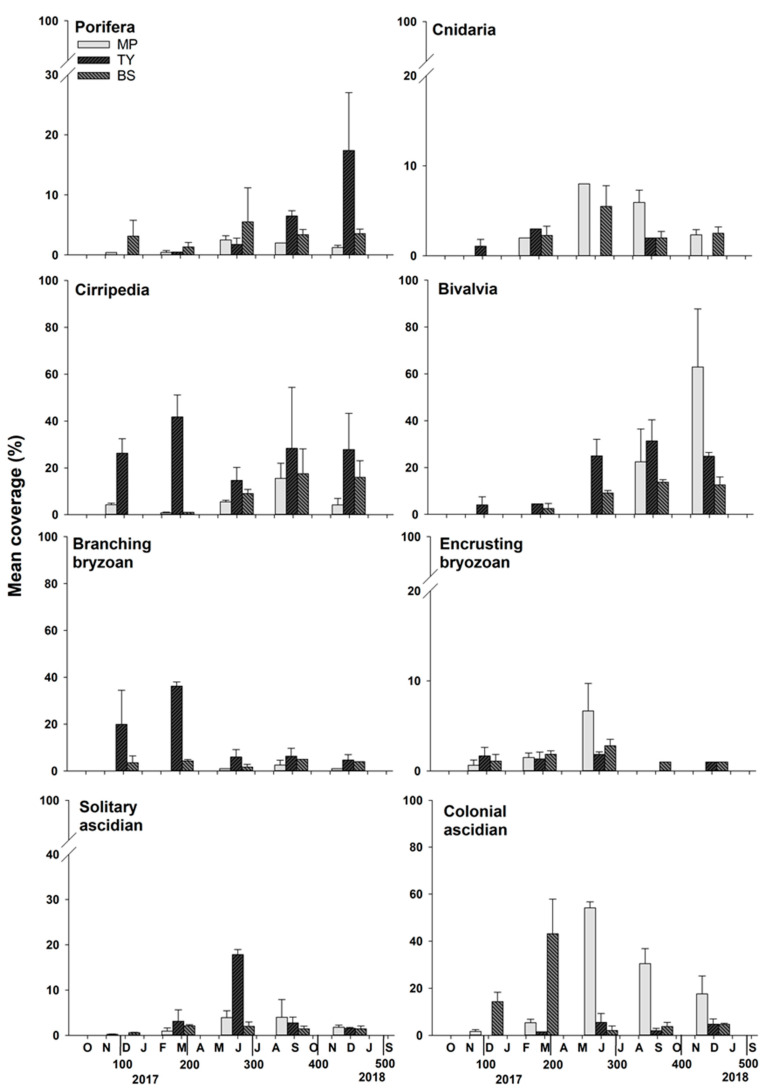
Results of mean coverage for eight taxa observed at the three sites during the study period.

**Figure 4 ijerph-19-01083-f004:**
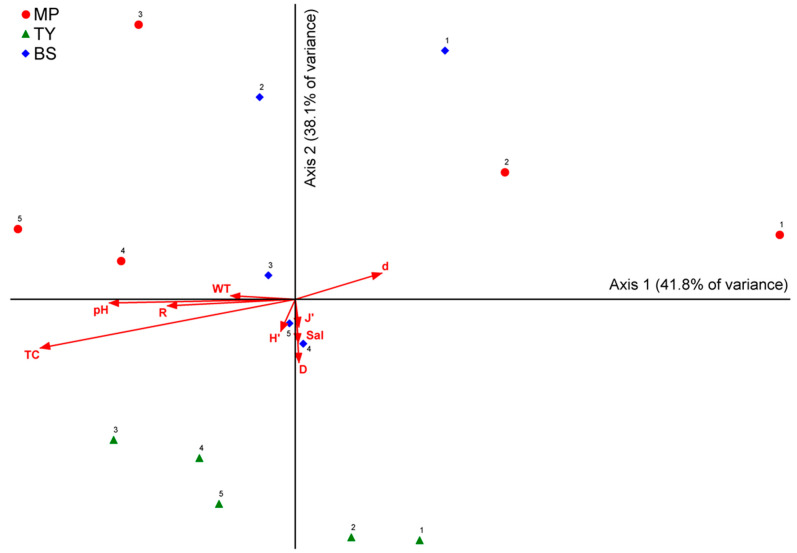
Results of the non-metric multidimensional scaling analysis showing sessile invertebrate communities (for taxa present in over 5% of the sample) at the survey sites during the study period. Number of each sites represents a time point. H’: diversity index, D: dominance index, J’: evenness index, d: richness index, R: species richness, TC: total coverage, WT: water temperature, Sal: salinity, and pH.

**Table 1 ijerph-19-01083-t001:** PERMANOVA results showing differences among the survey sites and substrates. The bold value denotes a significant result.

Variable	Mean Coverage (%)			
	df	MS	Pseudo-F	*p* (perm)
Site	2	33794	16.418	0.0001
Substrates	3	3253.7	1.5807	0.0615
Site × substrates	6	2413.2	1.1724	0.2253
Residual	150	2058.4		

**Table 2 ijerph-19-01083-t002:** Two-way ANOVA results for the types of artificial substrates and times using the mean coverage of taxa in each study site. Corrected *p* values were calculated using the Benjamini–Hochberg false discovery rate approach (*p* < 0.05).

Site	Taxonomic Group	Porifera		Cnidaria		Cirripedia		Bivalvia		Branching Bryozoan		Encrusting Bryozoan		Solitary Ascidian		Colonial Ascidian	
	Variable	F	*p*	F	*p*	F	*p*	F	*p*	F	*p*	F	*p*	F	*p*	F	*p*
MP	Times	3.19	0.08	2.41	0.22	**29.43**	**0.00**	8.91	0.01	0.54	0.91	2.32	0.28	1.30	0.52	**19.16**	**0.00**
	Substrates	0.67	0.77	2.07	0.28	**21.48**	**0.00**	0.53	0.87	0.15	0.96	0.30	0.94	0.25	0.95	**29.54**	**0.00**
	Times × substrates	0.54	0.92	0.88	0.76	**7.08**	**0.00**	1.02	0.67	0.24	0.96	1.21	0.59	0.90	0.76	**8.90**	**0.00**
TY	Times	**23.08**	**0.00**	1.00	0.74	**11.06**	**0.00**	**9.40**	**0.00**	**31.73**	**0.00**	0.92	0.73	**78.99**	**0.00**	3.42	0.09
	Substrates	2.87	0.14	0.37	0.90	2.29	0.23	**11.58**	**0.00**	0.75	0.76	2.12	0.26	2.14	0.25	0.46	0.90
	Times × substrates	1.73	0.22	0.37	0.96	0.38	0.96	2.63	0.11	0.54	0.95	2.04	0.19	0.97	0.72	0.35	0.96
BS	Times	1.61	0.36	1.49	0.47	**20.57**	**0.00**	**12.63**	**0.00**	2.56	0.18	0.53	0.88	**6.39**	**0.00**	**27.28**	**0.00**
	Substrates	1.52	0.41	0.39	0.90	**13.72**	**0.00**	**4.91**	**0.03**	0.41	0.89	2.32	0.22	1.93	0.28	2.59	0.19
	Times × substrates	0.54	0.93	0.64	0.89	**3.39**	**0.02**	1.25	0.55	0.84	0.79	0.98	0.73	0.93	0.77	2.23	0.11

Bold values denote significant results.

**Table 3 ijerph-19-01083-t003:** ANOSIM results comparing substrate types using the mean coverage of taxa in each site (R statistic > 0.5; *p* < 0.05).

Site	Substrate	Taxon	R Statistic	Significance Level (*p*)
MP	Stone vs. Rubber	Colonial ascidian	0.417	0.010
	**Stone vs. Tarpaulin**	Colonial ascidian	**0.692**	**0.008**
	**Stone vs. Iron**	Colonial ascidian	**0.521**	**0.011**
	Rubber vs. Tarpaulin	Cirripedia	0.203	0.009
	Rubber vs. Iron	Cirripedia	0.357	0.017
	Tarpaulin vs. Iron	Cnidaria	0.057	0.319
TY	Stone vs. Rubber	Bivalvia	0.393	0.056
	**Stone vs. Tarpaulin**	Bivalvia	**0.728**	**0.007**
	**Stone vs. Iron**	Bivalvia	**0.869**	**0.002**
	Rubber vs. Tarpaulin	Solitary ascidian	0.277	0.064
	Rubber vs. Iron	Branching bryozoan	0.207	0.088
	Tarpaulin vs. Iron	Cnidaria	0.500	0.200
BS	**Stone vs. Rubber**	Bivalvia	**0.517**	**0.001**
	**Stone vs. Tarpaulin**	Bivalvia	**0.619**	**0.005**
	**Stone vs. Iron**	Cirripedia	**0.745**	**0.011**
	Rubber vs. Tarpaulin	Branching bryozoan	0.361	0.044
	Rubber vs. Iron	Cirripedia	0.338	0.041
	Tarpaulin vs. Iron	Cirripedia	0.195	0.058

Bold values denote significant results.

## Data Availability

Data that support the findings of this study are available from the corresponding author upon reasonable request.

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
