# Peer review of "Comparison of Recruitment Patterns of Sessile Marine Invertebrates According to Substrate Characteristics"

_ijerph, 2022, doi:10.3390/ijerph19031083_

Round 1

Reviewer 1 Report

Biofouling is a major problem in coastal areas, and the importance of research themes is high. In addition, the reliability of the results is high by conducting field experiments using natural and artificial substrates in three sea areas for 16 months. In addition, there is no logical failure in the data analysis method. Therefore, this reviewer rates this paper as worthy of publication.
However, a major concern in the problem of bifouling is the possibility to select a substrate to which less organisms adhere. Therefore, data should be shown and explained whether the differences in the substrates used in the three sea areas affected the attached species and biomass.
On the other hand, different adherent species and biomass in different sea areas are expected results, and is it meaningful to analyze the interaction between sea areas and substrates? I think it is necessary to explain what the authors think about this.

Author Response

Biofouling is a major problem in coastal areas, and the importance of research themes is high. In addition, the reliability of the results is high by conducting field experiments using natural and artificial substrates in three sea areas for 16 months. In addition, there is no logical failure in the data analysis method. Therefore, this reviewer rates this paper as worthy of publication.

Answer:

  • We are grateful to the reviewer for insightful comments on my manuscript. We made changes to reflect most of the suggestions provided by the reviewer. We marked revisions with 'track changes' in the manuscript. Here is a point-by-point response to the reviewer’s comments and concerns, and all line numbers refer to the revised manuscript file with tracked changes.

However, a major concern in the problem of biofouling is the possibility to select a substrate to which less organisms adhere. Therefore, data should be shown and explained whether the differences in the substrates used in the three sea areas affected the attached species and biomass.

Answer:

  • We modified as suggested in results (line 234-239).

“When the coverage by major taxa was compared among substrates at each site, there was no significant difference in the community structure between natural and artificial substrate until about 200 Julian days. There was a difference in the rubber substrate of MP because Cirripedia had a higher abundance than other substrates, but there was no similar trend in other sites. However, the trend was similar to the two-way ANOVA result during the entire survey period (Table 2; Fig. S2).”

On the other hand, different adherent species and biomass in different sea areas are expected results, and is it meaningful to analyze the interaction between sea areas and substrates? I think it is necessary to explain what the authors think about this.

Answer:

  • As the reviewer pointed out, we also predicted that species and biomass would be different in different waters during the experimental design phase. However, in the study results, the tendency of the community structure was constant in the two sites (Mokpo and Busan) due to the influence of the substrate. Therefore, it is considered meaningful to analyze the interaction between the sea area and the substrate because the structure of the community was similar even if the sea area was different.

Reviewer 2 Report

I find the paper interestingly designed but with methodological (statistical) problems, and poorly presented. I believe that it will be worth publishing after addressing the issues listed below:

Abstract needs serious transformation as it is not very informative and very little results are presented there.

Line 13. The opening sentence in the abstract is obvious and I would suggest getting straight to the point/topic

Line 16. Please explain how the biofouling reduces the biodiversity as it is not clear?

Line 21. Avoid such general statements. Get to the point and provide how?, in which way? the trends were similar? In abundance, structure, dominance or biodiversity?

Line 23. “Significant trends”  in what??? Also not mentioned here…

Line 25-26 is also trivial, also there are no any antifouling methods described in the paper so why to mention them as an outcome of the paper?

Introduction

Line 31-32 Suggest to delete this sentence

Line 39-42 There is no mention on the presence of conspecifics. In the literature this is widely recognized factor. You will easily find many good references…

Line 75 “sea area differences” sounds vague. Consider “location” instead

Through the Introduction I have noticed several recent references, that could be helpful, are lacking. Please see e.g. paper by Witalis et al. from different harbours https://www.sciencedirect.com/science/article/abs/pii/S0272771420308386?casa_token=bpZCr7cn8vIAAAAA:k5mQ5p68bv-jDrDiYlBN-hFIb50VvWa4J8EqNhChmUoJxkF-W_lLXA1U4TMn5ixE1ejQj8I

And https://www.sciencedirect.com/science/article/pii/S0022098116301940?casa_token=ao_-wmoqqPIAAAAA:8psKxzPPMGVaWUzWzbJAn4uQ8Kj1bGaq7rT07RswM6uvFHu_pC58flcdmCZfkmkNL4sd8HIzbdk

And verify the factors that are mentioned therein…

Expand the hypotheses as at the moment they are very general. You could extract much more from your data.  In 2) you are expecting differences also in abundance, why not in hypothesis 1? Ad. 2) what kind of differences between the substrates you are talking about? Please specify.

Line 112 change “land” to “surface”

Methods

Line 137 describe how coverage was quantified?

Line 144 Instead of getting rid of rare species (do not deprive yourself of this valuable information) I would suggest keeping them and using a proper data transformation.

Multivariate analysis, should be accompanied not only with SIMPER but also with PERMDISP

Line 201-204 I think, as this is not clear, that you mistake PERMANOVA main test with pair-wise tests.. If there is no significant result from main test you do not test it further with pair-wise. Also when you perform  a pairwise test you indicate which substrate is differeing from the others…

Line 237 Higher from what?? Not clear

Discussion

Line 297-300 It is in fact a results section

Line 302-314 as above

Line 335-337 Cirripedia a globally distributed thus this statement sounds odd

Line 354-356 Rethink this sentence

Conclusions

Similary to abstract

Line 384 and 386 are repetitive

Line 389-390 Focuses only on environmental conditions without putting attention to interspecific conditions among taxa

Line 394-397 sounds trivial

Good luck!

Author Response

I find the paper interestingly designed but with methodological (statistical) problems, and poorly presented. I believe that it will be worth publishing after addressing the issues listed below:

Answer:

  • We are grateful to the reviewer for insightful comments on my manuscript. We made changes to reflect most of the suggestions provided by the reviewer. We marked revisions with 'track changes' in the manuscript. Here is a point-by-point response to the reviewer’s comments and concerns, and all line numbers refer to the revised manuscript file with tracked changes.

Abstract needs serious transformation as it is not very informative and very little results are presented there.

Line 13. The opening sentence in the abstract is obvious and I would suggest getting straight to the point/topic

Answer:

  • As suggested by the reviewer, the first sentence of the abstract presented the negative effects of marine invertebrates attached to artificial structures. (line 13-15)

Line 16. Please explain how the biofouling reduces the biodiversity as it is not clear?

Answer:

  • We corrected this sentence because it was considered to be an unclear explanation. (line 15-16).

Line 21. Avoid such general statements. Get to the point and provide how?, in which way? the trends were similar? In abundance, structure, dominance or biodiversity?

Answer:

  • We modified as suggested (line 21-22).

Line 23. “Significant trends” in what??? Also not mentioned here…

Answer:

  • We added a description (line 23)

Line 25-26 is also trivial, also there are no any antifouling methods described in the paper so why to mention them as an outcome of the paper?

Answer:

  • We deleted the description of antifouling that was not explained in the paper (line 25-26).

Introduction

Line 31-32 Suggest to delete this sentence

Answer:

  • We have deleted as suggested (line 31).

Line 39-42 There is no mention on the presence of conspecifics. In the literature this is widely recognized factor. You will easily find many good references…

Answer:

  • We modified as suggested in results (line 62-64).

Line 75 “sea area differences” sounds vague. Consider “location” instead

Through the Introduction I have noticed several recent references, that could be helpful, are lacking. Please see e.g. paper by Witalis et al. from different harbours https://www.sciencedirect.com/science/article/abs/pii/S0272771420308386?casa_token=bpZCr7cn8vIAAAAA:k5mQ5p68bv-jDrDiYlBN-hFIb50VvWa4J8EqNhChmUoJxkF-W_lLXA1U4TMn5ixE1ejQj8I

And https://www.sciencedirect.com/science/article/pii/S0022098116301940?casa_token=ao_-wmoqqPIAAAAA:8psKxzPPMGVaWUzWzbJAn4uQ8Kj1bGaq7rT07RswM6uvFHu_pC58flcdmCZfkmkNL4sd8HIzbdk

And verify the factors that are mentioned therein…

Answer:

  • We corrected the words by replacing 'sea area' with 'location' as mentioned.

Expand the hypotheses as at the moment they are very general. You could extract much more from your data.  In 2) you are expecting differences also in abundance, why not in hypothesis 1? Ad. 2) what kind of differences between the substrates you are talking about? Please specify.

Answer:

  • We corrected as suggested in Hypothesis 1), 'composition' was changed to 'structure' to include species abundance, and in Hypothesis 2), it was modified that the structure of a community affects the difference between natural and artificial substrates (line 73-76).

“1) differences in the community structure of natural substrates due to port location, 2) differences between natural and artificial substrates due to the community structure, and 3) effects of differences among artificial substrates on species composition and abundance.”

Line 112 change “land” to “surface”

Answer:

  • We modified as suggested (line 110-112).

Methods

Line 137 describe how coverage was quantified?

Answer:

  • We clarified how coverage was measured quantitatively (line 138-140).

Line 144 Instead of getting rid of rare species (do not deprive yourself of this valuable information) I would suggest keeping them and using a proper data transformation.

Answer:

  • Thank you for this suggestion. You have raised an important point here. However, removing species with an abundance of less than 5% is a method to increase the resolution of multivariate statistical analysis. If the rare species is a special species such as an invasive species, a separate explanation is provided. However, the rare species in our study did not have any special species. Therefore, in the multivariate analysis of this study, the analysis was performed after removing the species with less than 5% coverage, and detailed information on rare species can be found in Table 5.

Multivariate analysis, should be accompanied not only with SIMPER but also with PERMDISP

Answer:

  • Thank you for pointing this out. We agree with the reviewers that SIMPER and PERMDISP should be performed together in multivariate analysis. However, we performed ANOSIM from which the R value is derived to check how much the factor affects the variable, and the result is considered to be able to replace SIMPER and PERMDISP.

Line 201-204 I think, as this is not clear, that you mistake PERMANOVA main test with pair-wise tests.. If there is no significant result from main test you do not test it further with pair-wise. Also when you perform a pairwise test you indicate which substrate is differeing from the others…

Answer:

  • Thank you for this suggestion. You have raised an important point here. However, it appears to be slightly different from the purpose for PERMANOVA used in our study. PERMANOVA is a basic analysis that compares two factors in pair-wise, and since the analysis was performed according to the manual (Anderson et al., 2008), it is considered to be properly used. Therefore, we performed a pair-wise test on the factors, and as a result, we consider that there was a significant difference in site (factor1) but no significant difference in temperament (factor2)

Line 237 Higher from what?? Not clear

Answer:

  • We modified as suggested (line 245-246).

Discussion

Line 297-300 It is in fact a results section

Answer:

  • The following paragraph was modified (line 305-307).

“The analysis results of this study (nMDS and MRPP) based on benthic organisms of natural substrates also support this reasoning (Fig. 4; Table S4).”

Line 302-314 as above

Answer:

  • As your suggestion, we have revised the paragraph for the purpose of the discussion (line 309-313).

Line 335-337 Cirripedia a globally distributed thus this statement sounds odd

Answer:

  • Thanks for the suggestion and we agree. Since Cirripedia are distributed worldwide, the text has been modified to be ‘areas where Cirripedia dominates over other taxa' (line 304-306).

Line 354-356 Rethink this sentence

Answer:

  • Thank you for pointing this out. We agree with the reviewer's suggestion and we have deleted the sentence.
  •  

Conclusions

Similary to abstract

Line 384 and 386 are repetitive

Answer:

  • We have corrected the two sentences so that they are not repeated (line 393-394).

Line 389-390 Focuses only on environmental conditions without putting attention to interspecific conditions among taxa

Answer:

  • Thank you for this suggestion. You have raised an important point here. However, in our study, bivalves were the dominant taxon in TY. It is thought that the environment different from the other two regions had more influence than the interspecies interaction.

Line 394-397 sounds trivial

Answer:

  • We modified as suggested in conclusion (line 390-391).

Reviewer 3 Report

The manuscript submitted by Hui-Yun Huang  et al. “Comparison of Recruitment Patterns of Sessile Marine Inverte- 2 brates According to Substrate Characteristics” is well-written and discusses an interesting topic.

The main question addressed by the authors was: how artificial substrates affect the recruitment, abundance, and diversity of marine benthic invertebrates compared to natural substrates in several harbor habitats?

In my opinion, this investigation is original and has little attention in literature, and hence, it will provide very important recommendations, particularly for coastal countries, in the developing countries.

The conclusions are consistent with the evidence and arguments presented in the manuscript clearly answering the main question of the study.

Accordingly, I recommend accepting this paper in its current form despite its minor English language issues.

Author Response

The manuscript submitted by Hui-Yun Huang et al. “Comparison of Recruitment Patterns of Sessile Marine Inverte- 2 brates According to Substrate Characteristics” is well-written and discusses an interesting topic.

The main question addressed by the authors was: how artificial substrates affect the recruitment, abundance, and diversity of marine benthic invertebrates compared to natural substrates in several harbor habitats?

In my opinion, this investigation is original and has little attention in literature, and hence, it will provide very important recommendations, particularly for coastal countries, in the developing countries.

The conclusions are consistent with the evidence and arguments presented in the manuscript clearly answering the main question of the study.

Accordingly, I recommend accepting this paper in its current form despite its minor English language issues.

Answer:

  • Thank you for your valuable input in our manuscript. We have taken your suggestion and revised minor English language issues. We appreciate your positive thoughts and recommendation.

Reviewer 4 Report

This is a research report that belongs to basic biological sciences and is helpful to basic marine biological sciences. 1. In Table 2 why only 8 kinds of organisms are analyzed, can you explain the reasons? 2. The authors conducts the experiment in 3 places. I would like to ask if there is only one observation in each place or there are several points as an average. I did not see the explanation in the manuscript. I wish that the authors can illustrate it because it involves statistical significance. 3. On page 11, line 329, the authors mentioned that iron is corroded, which means that the iron ions in this area have increased. Is there any change in the algae population has been observed, because there have been studies published that increase the iron ions can increase the algae population . https://link.springer.com/content/pdf/10.1007/s10811-019-01915-5.pdf 4. The author's explanation for the figure 3 is not sufficient, I hope to add more specific explanation. 5. There are too many references and should be quoted with caution. I decided it should be a minor revision.

Author Response

This is a research report that belongs to basic biological sciences and is helpful to basic marine biological sciences.

Answer:

  • We are grateful to the reviewer for insightful comments on my manuscript. We made changes to reflect most of the suggestions provided by the reviewer. We marked revisions with 'track changes' in the manuscript. Here is a point-by-point response to the reviewer’s comments and concerns, and all line numbers refer to the revised manuscript file with tracked changes.

  1. In Table 2 why only 8 kinds of organisms are analyzed, can you explain the reasons?

Answer:

  • Thank you for pointing this out. We grouped them into eight taxa based on the morphological features that organisms attach to their substrates. This was because we considered that analyzing diversity in high taxa was appropriate to represent the characteristics of the substrate.

  1. The authors conducts the experiment in 3 places. I would like to ask if there is only one observation in each place or there are several points as an average. I did not see the explanation in the manuscript. I wish that the authors can illustrate it because it involves statistical significance.

Answer:

  • We conducted three replicates in one site. The average value of 3 replicates was used for the analysis, and materials and methods were modified to clarify this (line 101 and 151-152)

  1. On page 11, line 329, the authors mentioned that iron is corroded, which means that the iron ions in this area have increased. Is there any change in the algae population has been observed, because there have been studies published that increase the iron ions can increase the algae population. https://link.springer.com/content/pdf/10.1007/s10811-019-01915-5.pdf

Answer:

  • Thank you for this suggestion. We added the explanation (line 328-329).

  1. The author's explanation for the figure 3 is not sufficient, I hope to add more specific explanation.

Answer:

  • Thank you for pointing this out. Agreeing with the reviewer's comments, we added a detailed description of the figure 3 to the results and discussion (line 207-208, 209-210 and 304-307)

  1. There are too many references and should be quoted with caution. I decided it should be a minor revision.

Answer:

  • Thanks for the suggestion and we agree. As the reviewer advised, there are many references in this paper, so we were careful when citing. In addition, we have taken your suggestion and revised minor issues. We appreciate your positive thoughts and suggestion.

Reviewer 5 Report

Bae et al. compare the benthic macrofaunal settlement on four different substrates (one natural substrate -stone; three artificial substrates -rubber, tarpaulin, and iron) at three Korean harbour sites. The proportional coverage of the different invertebrate groups is monitored up to 16 months after the substrates had been installed. The authors discuss the different between sites and substrates and, based on their findings, provide several suggestions on how to prevent biofouling.

Overall, I liked reading the manuscript and I think the study was well-designed: the substrate selection is relevant and the replication across sites ensures robustness of the findings. The recommendations made with regard to biofouling strategies are nice and increase the relevance of the paper.

My only concern with this study is related to the statistics. I think this can largely be mediated by providing some additional information in the method section and perhaps adjusting/removing some of the tests. I discuss these issues below.

Line-by-line comments:

L13: characteristic of -> that includes

L14: have a negative effect on…. by attaching (specific on what they have a negative effect)

L16: ‘reduced biodiversity’ is vague and very context specific, I wouldn’t state that here

L21: differences from the other two sites

L31: invertebrates are comprised of various

L31: seaweeds are not invertebrates

L34: remove ‘these’

L34: ‘is generally biphasic’

L39: remove ‘in addition’

L43: negatively impact what?

L45: biofouling is a detrimental condition for what?

L61: according to -> in function of

L62: and the analysis of

L63: Recently,…carried out -> not a very informative sentences, I would include more details on this research instead (cf comment on this in the discussion)

L69: compared a variety of artificial substrates with natural substrates

L77: since you are comparing artificial substrates, I would add a third hypothesis on differences between artificial substrates specifically and potentially something on the interaction between site and substrate

L91 [important]: please specify throughout the manuscript where and how these three replicates per site are used. It is often not clear how these are treated in the different tests/if these are shown on the plots (eg. The nmds plot)

L102: I would add a comment on the fact that iron is rarely used as such as it is most often coated by e.g. paint

L110 [important]: please specify if the substrates where then return to the same position they were collected from (i.d. the same tiles are evaluated every three months) or the sampling is destructive and more tiles per substrates were places at each of the 3x3 locations

L136 [important]:here and later on as well: mean species rich/coverage -> averaged over the three replicates per site, averaged over time

L142 [important]: substrates as 2 levels (natural-artifical) or four levels

L151 [extremely important]: several of these ANOVAs were performed without taking the dependence of the time into consideration. As timepoints will not be independent of each other they should be treated as such, by e.g. using a repeated measures anova. In addition, none of the required assumptions of anova have been checked and no overall p-value correction (e.g. FDR) has been applied. As a result, I would suggest to be very carefull with these anova’s and ideally limit them to the species richness and just discuss the coverage different between groups/substrates/time points as such.

L142: the most conventional way to abbreviate Non-metric multidimensional scaling is as ‘NMDS’

L162&163: observed -> measured

L201: this is one of the cases where it is not directly clear what is tested (levels of substrate, what about the time points)

L202-206 [important]: this conflicts with the later findings and discussion. Please clarify this in the discussion. Why do you not observe significant differences here at the interaction between substrates and sites? You extensively discuss these differences later on, so either the variation is too large or differences are not that pronounced. Either way this needs to be addressed in the discussion

L207 & L217: please specify the diversity index, I’m assuming Shannon?

Fig.4 [important] : the number are the timepoints? Where are the replicates within site, are these already averaged? If so, please provide a plot were all points are shown

L227-227: time-dependence issue here, as well as multiple p-value correction and validity of the ANOVA assumptions

Table2: not all significant values are in bold and some of the MP Times cells are underlined

Table3: not all significant values are in bold

L255-265: I would move this part to either intro or method section

L267-272: I would reduce the amount of raw data that is provided since this is the discussion

L287-290: is this all related to the exposure time? I am assuming that the substrate were continuously submersed in which case the exposure is not relevant

L290-291: I think it is important to mention that these were Italian strains so they might be different in their response

L297-300: I would not show p-values here

L302-316: I think this text would fit better in the introduction

L339: I would immediately state what the difference was

L346-349: you have a lot of ‘however’ & ‘therefore’ in these sentences

L379: I think it would be good to include a critical note on the discovered site differences and on the limited time at which the coverage was evaluated. Both limit the scope for which the provided management solutions could be applied (substrates of structures will be in the water for much longer and the strong differences between sites suggest that a site based approach might be best to optimise local biofouling strategies)

Author Response

Bae et al. compare the benthic macrofaunal settlement on four different substrates (one natural substrate -stone; three artificial substrates -rubber, tarpaulin, and iron) at three Korean harbour sites. The proportional coverage of the different invertebrate groups is monitored up to 16 months after the substrates had been installed. The authors discuss the different between sites and substrates and, based on their findings, provide several suggestions on how to prevent biofouling.

Overall, I liked reading the manuscript and I think the study was well-designed: the substrate selection is relevant and the replication across sites ensures robustness of the findings. The recommendations made with regard to biofouling strategies are nice and increase the relevance of the paper.

My only concern with this study is related to the statistics. I think this can largely be mediated by providing some additional information in the method section and perhaps adjusting/removing some of the tests. I discuss these issues below.

Answer:

  • We are grateful to the reviewer for insightful comments on my manuscript. We made changes to reflect most of the suggestions provided by the reviewer. We marked revisions with 'track changes' in the manuscript. Here is a point-by-point response to the reviewer’s comments and concerns, and all line numbers refer to the revised manuscript file with tracked changes.

Line-by-line comments:

L13: characteristic of -> that includes

Answer:

  • We modified as suggested (line 13).

L14: have a negative effect on…. by attaching (specific on what they have a negative effect)

Answer:

  • We modified as suggested (line 13-15).

L16: ‘reduced biodiversity’ is vague and very context specific, I wouldn’t state that here

Answer:

  • We changed as suggested and deleted (line 15-16).

L21: differences from the other two sites

Answer:

  • We changed as suggested (line 21-22).

L31: invertebrates are comprised of various

L31: seaweeds are not invertebrates

Answer:

  • Thank you for pointing this out. In response to the comments of other reviewers, we have deleted this sentence.

L34: remove ‘these’

L34: ‘is generally biphasic’

L39: remove ‘in addition’

Answer:

  • We changed as suggested and deleted (line 34-39).

L43: negatively impact what?

Answer:

  • We added an explanation (line 40-42).

L45: biofouling is a detrimental condition for what?

Answer:

  • We corrected the sentences (line 44-45).

L61: according to -> in function of

L62: and the analysis of

Answer:

  • We modified as suggested (line 61-62).

L63: Recently,…carried out -> not a very informative sentences, I would include more details on this research instead (cf comment on this in the discussion)

Answer:

  • We revised the contents of the discussion and added it to the introduction (line 62-64).

L69: compared a variety of artificial substrates with natural substrates

Answer:

  • We modified as suggested (line 67-68).

L77: since you are comparing artificial substrates, I would add a third hypothesis on differences between artificial substrates specifically and potentially something on the interaction between site and substrate

Answer:

  • Thank you for these detailed comments. We added the hypothesis as suggested (line 74-76).

L91 [important]: please specify throughout the manuscript where and how these three replicates per site are used. It is often not clear how these are treated in the different tests/if these are shown on the plots (eg. The nmds plot)

Answer:

  • As you suggested, we clarified replicates in materials and methods (line 89 and 137)

L102: I would add a comment on the fact that iron is rarely used as such as it is most often coated by e.g. paint

Answer:

  • We added as suggested (line 101-103).

L110 [important]: please specify if the substrates where then return to the same position they were collected from (i.d. the same tiles are evaluated every three months) or the sampling is destructive and more tiles per substrates were places at each of the 3x3 locations

Answer:

  • We added a detailed description of materials and methods (line 113-114).

L136 [important]: here and later on as well: mean species rich/coverage -> averaged over the three replicates per site, averaged over time

Answer:

  • We modified as suggested (line 137-138).

L142 [important]: substrates as 2 levels (natural-artifical) or four levels

L142: the most conventional way to abbreviate Non-metric multidimensional scaling is as ‘NMDS’

Answer:

  • We changed as suggested (line 142-143).

L151 [extremely important]: several of these ANOVAs were performed without taking the dependence of the time into consideration. As timepoints will not be independent of each other they should be treated as such, by e.g. using a repeated measures anova. In addition, none of the required assumptions of anova have been checked and no overall p-value correction (e.g. FDR) has been applied. As a result, I would suggest to be very carefull with these anova’s and ideally limit them to the species richness and just discuss the coverage different between groups/substrates/time points as such.

Answer:

  • Thank you for these detailed comments. As the reviewer pointed out, we performed the analysis without considering the time point in a two-way ANOVA. However, two-way ANOVA was a procedure to determine which of the two factors (site and time) influences the taxa in the overall study results. Repeated measures ANOVA considering time point can be confirmed in Supplementary Figure S2.

L162&163: observed -> measured

Answer:

  • Thank you for this suggestion. You have raised an important point here. Unfortunately, it was expressed that it was observed because the value measured by another institution was used, not the value measured directly by us.

L201: this is one of the cases where it is not directly clear what is tested (levels of substrate, what about the time points)

Answer:

  • We corrected that to be clearly expressed about the substrate and time points (line 204-205).

L202-206 [important]: this conflicts with the later findings and discussion. Please clarify this in the discussion. Why do you not observe significant differences here at the interaction between substrates and sites? You extensively discuss these differences later on, so either the variation is too large or differences are not that pronounced. Either way this needs to be addressed in the discussion

Answer:

  • Thank you for this suggestion. As you pointed out, the reason why there was no difference between the substrate and site is described in the discussion (line 258-280), and the contents of Table 1 were clarified, and description was added (line 281-282).

L207 & L217: please specify the diversity index, I’m assuming Shannon?

Answer:

  • As you pointed out, we have edited to be clear about the types of ecological indices, including diversity indices (line 209-213).

Fig.4 [important] : the number are the timepoints? Where are the replicates within site, are these already averaged? If so, please provide a plot were all points are shown

Answer:

  • Since numbers are time points, clarified the expression that the number represents the time point in the legend. All values used in the analysis are averages of three replicates, so there is one plot per time point in nMDS (Figure 4).

L227-227: time-dependence issue here, as well as multiple p-value correction and validity of the ANOVA assumptions

Answer:

  • Thanks to the reviewer for pointing out the p value correction. This part was not considered by us, and the p value in Table 2 was corrected using the Hochberg-Bonferroni method. A graph comparing raw p values and Hochberg-Bonferroni values can be seen in Supplementary Figure S3.

Table2: not all significant values are in bold and some of the MP Times cells are underlined

Answer:

  • We modified Table 2

Table3: not all significant values are in bold

Answer:

  • We modified Table 3

L255-265: I would move this part to either intro or method section

Answer:

  • Thanks for your suggestion, we moved that paragraph to materials and methods (line 89-93).

L267-272: I would reduce the amount of raw data that is provided since this is the discussion

Answer:

  • We changed as suggested (line 276-278).

L287-290: is this all related to the exposure time? I am assuming that the substrate were continuously submersed in which case the exposure is not relevant

Answer:

  • As you pointed out, this sentence was deleted because the substrate was always submerged regardless of the tide.

L290-291: I think it is important to mention that these were Italian strains so they might be different in their response

Answer:

  • Thanks for your point out. We cited Brunetti et al. (1980) in our paper because Italy is located at a similar latitude as the study site (latitude 30-35°), and the results of previous studies (Brunetti et al., 1980) are also consistent with the environment of the study area (line 224 – 225).

L297-300: I would not show p-values here

Answer:

  • As you suggested, we modified (line 305-307).

L302-316: I think this text would fit better in the introduction

Answer:

  • We modified as suggested (line 62-64).

L339: I would immediately state what the difference was

L346-349: you have a lot of ‘however’ & ‘therefore’ in these sentences

Answer:

  • We modified as suggested (line 349-350 and 359-361).

L379: I think it would be good to include a critical note on the discovered site differences and on the limited time at which the coverage was evaluated. Both limit the scope for which the provided management solutions could be applied (substrates of structures will be in the water for much longer and the strong differences between sites suggest that a site based approach might be best to optimise local biofouling strategies)

Answer:

  • Thanks for the suggestion and we agree. The period we conducted the survey was limited, and the structures in each survey site may have been submerged for a longer period of time. Therefore, an additional critical view of the above facts is included in the conclusion (line 405-410).

Round 2

Reviewer 2 Report

I am often forced to refer to previous review, thus I have highlighted my current concerns with blue colour for clarity.

The opening sentence in the abstract, is still obvious - adding the words "that includes" is not changing its meaning at all - it is still the same sentence, right? Instead I would start with the second, now adjusted, one and correct the word "community" into "communities", and most importantly describe/name what kind of communities are you writing about? Sessile, epifaunal, benthic invertebrate, larval? Would recommend being more specific, here, and in the whole abstract (as stated previously, unfortunately with little understanding) providing more details, actual results (instead of saying "there were differences") as this is abstract of the whole paper and the reader will decide whether to go on further or skip it... Perhaps it would be a good idea to join sentence 2 and 3 as now they are describing the same thing, but poorly (e.g.  are the operating costs are the only cause? or the most important one?)

Line 21. Avoid such general statements. Get to the point and provide how?, in which way? the trends were similar? In abundance, structure, dominance or biodiversity?

Answer: We modified as suggested (line 21-22).

The modification proposed by the authors does not change anything. Similiar community structure in terms of what? Dominance? Species pool? Diversity??? The same kind of question applies to differences in Tongyeong.

Line 23. “Significant trends” in what??? Also not mentioned here…

Answer: We added a description (line 23)

Where? You did not (!)

The last sentence of the article remains trival as it was ("We recommend avoiding substrates that potentially promote biological contamination for antifouling and management of biofouling") Is this really the most important outcome of this paper?

I also find the the most important remark that abstract needs serious transformation as it is not very informative and very little results are presented there, was unfortunately not taken seriously.

Expand the hypotheses as at the moment they are very general. You could extract much more from your data. In 2) you are expecting differences also in abundance, why not in hypothesis 1? Ad. 2) what kind of differences between the substrates you are talking about? Please specify.

The hypothesis two 2) differences between natural and artificial substrates due to the community structure, is unclear. Did you mean local community structure? Do you quantify it in the study???

Line 144 Instead of getting rid of rare species (do not deprive yourself of this valuable information) I would suggest keeping them and using a proper data transformation.

I am very surprised to see that removing species is a method "to increase the resolution of multivariate statistical analysis" (!)

Multivariate analysis, should be accompanied not only with SIMPER but also with PERMDISP.

Answer: Thank you for pointing this out. We agree with the reviewers that SIMPER and PERMDISP should be performed together in multivariate analysis. However, we performed ANOSIM from which the R value is derived to check how much the factor affects the variable, and the result is considered to be able to replace SIMPER and PERMDISP.

This answer convinced me that the authors have a very little idea what they are writing about. ANOSIM is not a test that could replace any of this analyses. It provides an estimation on how the communities are differing from each other (and is ran on ranks in contradiction to PERMANOVA), while SIMPER gives an indication what kind of species are driving this differences. Most importantly, PERMDISP is checking the difference in dispersion between the groups. It can identify if it is the dispersion of the group that is driving the significance of the PERMANOVA test or if it is the differences of the group data themselves. In other words, it may be possible that your PERMANOVA results are wrong... and thus the conlusions.

Line 201-204 I think, as this is not clear, that you mistake PERMANOVA main test with pair-wise tests.. If there is no significant result from main test you do not test it further with pair-wise. Also when you perform a pairwise test you indicate which substrate is differeing from the others…

Answer: Thank you for this suggestion. You have raised an important point here. However, it appears to be slightly different from the purpose for PERMANOVA used in our study. PERMANOVA is a basic analysis that compares two factors in pair-wise, and since the analysis was performed according to the manual (Anderson et al., 2008), it is considered to be properly used. Therefore, we performed a pair-wise test on the factors, and as a result, we consider that there was a significant difference in site (factor1) but no significant difference in temperament (factor2)

It is not clear to me. Especially that in the lines 213-215 you have a statement that contradicts the results form the Table 1 (!) Please provide also the table results for PERMANOVA pair-wise comparison between the sites. At the current version of the manuscript you provide the results of the main test only (Table 1).

Line 237 Higher from what?? Not clear

Answer: We modified as suggested (line 245-246).

It would be good to indicated whcih species are responisble for the observed differences with SIMPER analysis, and a table...

Author Response

The opening sentence in the abstract, is still obvious - adding the words "that includes" is not changing its meaning at all - it is still the same sentence, right? Instead I would start with the second, now adjusted, one and correct the word "community" into "communities", and most importantly describe/name what kind of communities are you writing about? Sessile, epifaunal, benthic invertebrate, larval? Would recommend being more specific, here, and in the whole abstract (as stated previously, unfortunately with little understanding) providing more details, actual results (instead of saying "there were differences") as this is abstract of the whole paper and the reader will decide whether to go on further or skip it... Perhaps it would be a good idea to join sentence 2 and 3 as now they are describing the same thing, but poorly (e.g. are the operating costs are the only cause? or the most important one?)

Answer:

We are grateful to the reviewer for insightful comments on my manuscript. We made changes to reflect most of the suggestions provided by the reviewer. We marked revisions with 'track changes' in the manuscript. Here is a point-by-point response to the reviewer’s comments and concerns, and all line numbers refer to the revised manuscript file with tracked changes.

We modified the abstract as suggested.

Line 21. Avoid such general statements. Get to the point and provide how?, in which way? the trends were similar? In abundance, structure, dominance or biodiversity?

Answer: We modified as suggested (line 21-22).

The modification proposed by the authors does not change anything. Similiar community structure in terms of what? Dominance? Species pool? Diversity??? The same kind of question applies to differences in Tongyeong.

Answer:

We modified as suggested (line 22-29).

Line 23. “Significant trends” in what??? Also not mentioned here…

Answer: We added a description (line 23)

Where? You did not (!)

The last sentence of the article remains trival as it was ("We recommend avoiding substrates that potentially promote biological contamination for antifouling and management of biofouling") Is this really the most important outcome of this paper?

I also find the the most important remark that abstract needs serious transformation as it is not very informative and very little results are presented there, was unfortunately not taken seriously.

Answer:

We modified the abstract as suggested.

Expand the hypotheses as at the moment they are very general. You could extract much more from your data. In 2) you are expecting differences also in abundance, why not in hypothesis 1? Ad. 2) what kind of differences between the substrates you are talking about? Please specify.

The hypothesis two 2) differences between natural and artificial substrates due to the community structure, is unclear. Did you mean local community structure? Do you quantify it in the study???

Answer:

We modified as suggested (line 81-84).

Line 144 Instead of getting rid of rare species (do not deprive yourself of this valuable information) I would suggest keeping them and using a proper data transformation.

I am very surprised to see that removing species is a method "to increase the resolution of multivariate statistical analysis" (!)

Answer:

Thank you for this suggestion. You have raised an important point here. The need for careful treatment of rare species in multivariate analysis is a topic that has been continuously debated. Researchers either transform or remove rare species data according to their respective purposes. Therefore, we decided to eliminate the rare species according to the guidelines in the PC-ORD manual (McCune and Grace, 2002) and with reference to other studies (Kim et al., 2018).

Kim, D.G.; Yoon, T.J.; Baek, M.J.; Bae, Y.J. Impact of Rainfall Intensity on Benthic Macroinvertebrate Communities in a Mountain Stream under the East Asian Monsoon Climate. J Freshwater Ecol 2018, 33, 489–501, doi:10.1080/02705060.2018.1476271.

Multivariate analysis, should be accompanied not only with SIMPER but also with PERMDISP.

Answer: Thank you for pointing this out. We agree with the reviewers that SIMPER and PERMDISP should be performed together in multivariate analysis. However, we performed ANOSIM from which the R value is derived to check how much the factor affects the variable, and the result is considered to be able to replace SIMPER and PERMDISP.

This answer convinced me that the authors have a very little idea what they are writing about. ANOSIM is not a test that could replace any of this analyses. It provides an estimation on how the communities are differing from each other (and is ran on ranks in contradiction to PERMANOVA), while SIMPER gives an indication what kind of species are driving this differences. Most importantly, PERMDISP is checking the difference in dispersion between the groups. It can identify if it is the dispersion of the group that is driving the significance of the PERMANOVA test or if it is the differences of the group data themselves. In other words, it may be possible that your PERMANOVA results are wrong... and thus the conlusions.

Answer:

We modified as suggested (line 227-233; Table S5 and S6).

Line 201-204 I think, as this is not clear, that you mistake PERMANOVA main test with pair-wise tests.. If there is no significant result from main test you do not test it further with pair-wise. Also when you perform a pairwise test you indicate which substrate is differeing from the others…

Answer: Thank you for this suggestion. You have raised an important point here. However, it appears to be slightly different from the purpose for PERMANOVA used in our study. PERMANOVA is a basic analysis that compares two factors in pair-wise, and since the analysis was performed according to the manual (Anderson et al., 2008), it is considered to be properly used. Therefore, we performed a pair-wise test on the factors, and as a result, we consider that there was a significant difference in site (factor1) but no significant difference in temperament (factor2)

It is not clear to me. Especially that in the lines 213-215 you have a statement that contradicts the results form the Table 1 (!) Please provide also the table results for PERMANOVA pair-wise comparison between the sites. At the current version of the manuscript you provide the results of the main test only (Table 1).

Answer:

We modified as suggested (line 227-233; Table S5).

Line 237 Higher from what?? Not clear

Answer: We modified as suggested (line 245-246).

It would be good to indicated whcih species are responisble for the observed differences with SIMPER analysis, and a table...

Answer:

We modified as suggested (line 282-288; Table S6).